# Optimizing Resources for On-the-Fly Label Estimation with Multiple Unknown Medical Experts

Tim Bary[1,*], Tiffanie Godelaine[1], Axel Abels[2,3,4], Benoît Macq[1]

[1]ICTEAM, UCLouvain, Louvain-la-Neuve, Belgium

[2]Machine Learning Group, Université Libre de Bruxelles, Brussels, Belgium

[3]AI Lab, Vrije Universiteit Brussel, Brussels, Belgium

[4]FARI Institute, Université Libre de Bruxelles - Vrije Universiteit Brussel, Brussels, Belgium

[*]Corresponding author: tim.bary@uclouvain.be

*Abstract*—Accurate ground truth estimation in medical screening programs often relies on coalitions of experts and peer second opinions. Algorithms that efficiently aggregate noisy annotations can enhance screening workflows, particularly when data arrive continuously and expert proficiency is initially unknown. However, existing algorithms do not meet the requirements for seamless integration into screening pipelines. We therefore propose an adaptive approach for real-time annotation that (I) supports on-the-fly labeling of incoming data, (II) operates without prior knowledge of medical experts or pre-labeled data, and (III) dynamically queries additional experts based on the latent difficulty of each instance. The method incrementally gathers expert opinions until a confidence threshold is met, providing accurate labels with reduced annotation overhead. We evaluate our approach on three multi-annotator classification datasets across different modalities. Results show that our adaptive querying strategy reduces the number of expert queries by up to **50%** while achieving accuracy comparable to a non-adaptive baseline. Our code is available at https://github.com/tbary/MEDICS.

*Index Terms*—Ground Truth Inference, Computer-in-the-Loop Diagnosis, Decision Making in Medicine, AI for Medical Diagnosis.

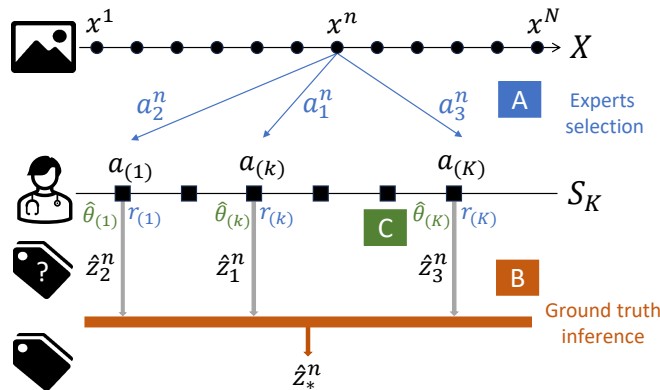

Fig. 1. Illustration of the problem setup. Given a data point $x^n$, the algorithm must (A) select $Q^n$ experts $\{a_q\}_{q=1}^{Q^n}$ from a coalition of size $K$ to annotate $x^n$, (B) infer a coalitional label $\hat{z}_*^n$ based on the experts' estimated labels $\{\hat{z}_q^n\}_{q=1}^{Q^n}$, and (C) update the estimated parameters $\{\hat{\theta}_{(k)}\}_{k=1}^{K}$ and trusts $\{r_{(k)}\}_{k=1}^{K}$ of the experts. In this example, $Q^n = 3$.

## I. INTRODUCTION

Estimating ground truths in medical screening programs involves significant challenges due to heterogeneity in expert opinions. Such variations in interpretations among healthcare professionals typically arise from differences in experience, training, and subjective judgment. This variability hinders ground truth inference, which in turn can result in inconsistent diagnoses, unnecessary and costly procedures, or untreated patients [1].

Aggregating annotations from multiple experts offers a way to mitigate these inconsistencies. Approaches inspired by crowdsourcing aim to integrate noisy or biased expert labels into more reliable consensus decisions [2]. When combined with active selection schemes that allocate expert attention based on data difficulty, these algorithms could substantially enhance clinical decision-making processes. However, existing methods do not fully align with current medical practices, making their deployment into real-time medical workflows challenging.

In screening programs, practitioners typically face a continuous stream of cases that require timely annotation decisions. Labeling involves progressively gathering opinions from multiple experts until sufficient confidence is reached [3]. This process not only supports clinical decisions but also contributes to peer learning and quality assurance [4]. Given the limited availability of medical experts, it is essential to allocate annotation effort efficiently, that is, ensuring that more difficult or ambiguous cases receive greater attention [5], [6].

Most crowdsourcing-derived algorithms differ from this paradigm by at least one of three ways. (I) Some assume that all data are available upfront, making it easier to pre-allocate annotators to difficult cases. (II) Others require prior knowledge of expert performance, labeled training data, or ground truth feedback—resources that are unavailable or costly to obtain in real-time medical settings. (III) Finally, approaches compatible with streaming data typically focus on selecting a single annotator, rather than building consensus from multiple expert inputs.

In this paper, we build on established elements from the crowdsourcing literature to propose an adaptive, stream-based method for real-time annotation that addresses a contextual gap in applying such techniques to medical workflows. The method (I) supports **on-the-fly labeling** of incoming data,

(II) operates **without prior knowledge** of expert capabilities or pre-labeled data, and (III) **dynamically allocates** additional experts based on the latent difficulty of each instance. It incrementally collects expert opinions until a predefined confidence level is reached, ensuring a proper distribution of the annotation resources.

We evaluate this approach on three multi-expert datasets with known ground truths and distinct data modalities. Results show that our adaptive querying reduces the number of expert queries by up to 50% while achieving accuracy comparable to a non-adaptive baseline.

To encourage reproducibility, our code is made available on GitHub.

## II. RELATED WORKS

We identify three key properties required for the integration of ground truth inference algorithms into real-time labeling workflows. These are: (I) On-the-fly annotation of data from a stream, (II) Possibility to cold start the algorithm on new medical expert groups without requiring external information (*i.e.*, prior knowledge on practitioners quality, labeled pre-training data, or ground truth feedback), and (III) Allocation of more resources to label the most uncertain data.

We categorize previous research algorithms based on their adherence to these three properties, and summarize this discussion in Table I. For algorithms that comply with all three properties, we clarify what other shortcomings they might have regarding annotations in medical settings. Works that lack all properties are excluded from our discussion.

### A. Algorithms with property (I)

Guan *et al.* [7] train a model along with multiple independent decision heads that each capture the behavior of one expert. After training, the model simulates the experts by issuing pseudo-annotations on unseen data and selects one via weighted majority voting. Keswani *et al.* [8] jointly learn a classifier and a deferral system. Once trained, the deferral system elects committees of human and artificial (*i.e.*, models) experts to label unseen data.

On top of requiring training data, these two algorithms are also coalition-specific. This means that, upon the departure or arrival of an expert from the coalition, the model must be trained anew.

Abels *et al.* [9] follow a contextual multi-armed bandits approach to identify consistent biases in experts behaviors, enhancing decisions accuracy. However, this strategy requires ground truth feedback after each decision, which is not always available.

### B. Algorithms with property (II)

Raykar *et al.* [10] and Rodrigues *et al.* [11] apply the Expectation-Maximization (EM) algorithm [22] to multi-annotated data to jointly estimate ground truth labels, annotator reliabilities, and inference model weights. Rodrigues *et al.*outperform Raykar *et al.*by modeling annotator reliabilities as latent variables, rather than the ground truths.

TABLE I
PROPERTIES OF THE ALGORITHMS FROM THE RELATED WORKS.
(I) ON-THE-FLY ANNOTATION OF DATA FROM A STREAM,
(II) POSSIBILITY TO COLD START THE ALGORITHM ON NEW COALITION
WITHOUT REQUIRING EXTERNAL INFORMATION, AND (III) ALLOCATION
OF MORE RESOURCES TO LABEL THE MOST UNCERTAIN DATA.

|  | [7]–[9] | [10]–[14] | [15], [16] | [17], [18] | [19] | [20], [21] | Ours |
|---|---|---|---|---|---|---|---|
| **(I)** | ✓ | | ✓ | | ✓ | ✓ | ✓ |
| **(II)** | | ✓ | ✓ | ✓ | | ✓ | ✓ |
| **(III)** | | | | ✓ | ✓ | ✓ | ✓ |

Later, Rodrigues *et al.* [12] use Gaussian Processes to infer ground truths and annotator reliabilities, integrating this into an active learning framework to select both the next data point and the most suitable annotator.

Similarly, Yan *et al.* [13] train a model in an active learning setup, retrospectively selecting the best image-annotator pair by assuming each expert is reliable on a specific data subset. Once the subset is identified, the expert most suited to it is queried.

Yu *et al.* [14] address multi-label datasets (*e.g.*, multiple pathologies per patient) by clustering annotators and selecting sample-label-worker triplets based on uncertainty, estimated information gain, and worker credibility.

### C. Algorithms with properties (I) and (II)

Donmez *et al.* [15] employ interval estimation to identify trustworthy experts within a coalition for on-the-fly data annotation. The trust in an expert is determined by the general agreement between its annotations and the majority decision.

Gimelfarb *et al.* [16] presents a Bayesian Model Combination method for reinforcement learning that adaptively learns to combine multiple expert reward shaping functions during training. This approach speeds up convergence and preserves optimal policies, all without adding runtime complexity.

### D. Algorithms with properties (II) and (III)

Welinder *et al.* [17] propose a two-step algorithm relying on EM and performed on a growing pool of images. In the label collection step, images receive labels from both approved and unknown experts until a confidence or a cost threshold is reached. In the second step, part of the unknown experts are approved based on the estimated quality of their labels.

Chen *et al.* [18] leverage a Markov Decision Process to select which data in a pool should be annotated next until a budget is exhausted. Annotated data are put back into the pool for potential re-annotation by other experts.

### E. Algorithm with properties (I) and (III)

Nguyen *et al.* [19] introduce CLARA, a system combining a Bayesian model with predictions from several machine learning models for more efficient multi-expert labeling. Tested on tasks aimed at flagging guideline-violating content, this approach requires substantial training data with known ground truth for accurate predictions.

## F. Algorithms with properties (I), (II), and (III)

Abraham *et al.* [20] group experts into crowds with common response distributions, using a multi-armed bandit algorithm to sample annotations until a stopping criterion is met. Though it satisfies all three properties, this approach is impractical for medical applications, as it requires large, homogeneous medical expert groups, which are rare in this field.

In a subsequent work, Abraham *et al.* [21] propose an algorithm that selects experts based on task difficulty. The paper offers two weighting approaches, depending on whether prior information about the experts' skills is available. When experts' skills are partially known, their decisions are weighted accordingly. Because knowledge about the experts is required in advance, this approach does not satisfy property (II). For coalitions with unknown skill levels, a majority vote is used. This latter approach could however be enhanced, as crowd-sourcing studies show that unweighted votes are less accurate than weighted ones [9], [10], [17], [18].

In summary, existing approaches do not meet the demands of real-time medical annotation, where prior knowledge is scarce and difficult cases require more attention. Addressing all three aspects—stream-based labeling, cold-start capability, and adaptive expert allocation——is therefore essential for building practical annotation systems in clinical settings.

## III. PROBLEM STATEMENT

Let $\boldsymbol{X} = \{x^n\}_{n=1}^N$ denote a stream of $N$ data points, where each $x^n$ has a unique hidden true label $z^n$, and let $S_K = \{a_{(k)}\}_{k=1}^K$ be a coalition of $K$ experts. The algorithm has, for each $x^n$, to query $Q^n \in [1, K]$ experts $\{a_q\}_{q=1}^{Q^n} \subseteq S_K^1$, and collect $\{\hat{z}_q^n\}_{q=1}^{Q^n}$, their estimated labels for $x^n$. The algorithm must then aggregate the obtained labels into a coalitional label $\hat{z}_*^n$ so that $\frac{1}{N} \sum_{n=1}^N \mathbb{1}_{\hat{z}_*^n = z^n}$ is maximized, where $\mathbb{1}$ is the indicator function. The problem is illustrated in Fig. 1.

Each expert $a_{(k)}$ can be characterized by $\hat{\boldsymbol{\theta}}_{(k)} = \{t_{(k)}, s_{(k)}\}$, where $t_{(k)}$ is the number of times the expert was queried, and $s_{(k)}$ is the estimated number successful answers they provided. These parameters determine a trust value $r_{(k)}$. Although richer experts models (*e.g.*, confusion matrices) could be used, these would increase with the number of classes and therefore require much more samples for precise expert evaluation [23].

We define the labeling cost as $C := \frac{1}{N} \sum_{n=1}^N Q^n$, that is, the average number of expert queries per data point, assuming unit cost per query. We resorted to this simplification because modeling uneven or varying expert costs is difficult without making unverifiable assumptions. It is however possible to add query cost as a third expert parameter. Since each data point must be labeled at least once, we have that $C \geq 1$. We do not impose a per-expert query limit, so the cost may be unevenly distributed among experts.

---

[1] Note that $(k)$ indexes experts in the coalition, while $q$ reflects the order in which the algorithm queries them.

## IV. ALGORITHM

This section introduces the adaptive algorithm developed in this work. It is designed as a backbone that incorporates three abstract functions, which allow for customized implementations tailored to specific task requirements. Our particular implementation of these functions is detailed later in the section.

### A. Backbone

To handle data points that exhibit varying, *a priori* unknown, levels of difficulty, our algorithm assesses whether additional expert opinions are needed by assigning a confidence level $c_*^n \in [0, 1]$ to the current coalitional label $\hat{z}_*^n$. Further expert input is requested only when the confidence in the decision remains low; that is, if $c_*^n$ does not meet a predefined threshold $\tau$. This threshold enables the user to adjust the trade-off between labeling accuracy and cost. Lowering $\tau$ favors cost savings but may reduce accuracy, whereas increasing it raises both the cost and accuracy of the labels.

Our algorithm (Algorithm 1) proceeds as follows: for each data point $x^n$, a first abstract function $\mathcal{A}$ calculates the trust levels $\{r_{(k)}\}_{k=1}^K$ of experts in the coalition $S_K$ and orders them. The two highest-ranked experts, $a_1$ and $a_2$, are queried to annotate $x^n$, resulting in labels $\hat{z}_1^n$ and $\hat{z}_2^n$, respectively. The second function, $\mathcal{B}$, infers a coalitional label $\hat{z}_*^n$ and a confidence measure $c_*^n$ based on the experts' responses $\hat{\mathbf{z}}^n = \{\hat{z}_1^n, \hat{z}_2^n\}$ and trusts $\mathbf{r}^n = \{r_1, r_2\}$.

If $c_*^n$ exceeds $\tau$, the algorithm deems the confidence in the label sufficient and submits it as the final output. Otherwise, as long as the threshold remains unmet and unqueried experts are available, the algorithm queries the highest-ranked unqueried expert $a_q$. It then recalculates $\hat{z}_*^n$ and $c_*^n$ by calling $\mathcal{B}$ on the updated sets of queried labels $\hat{\mathbf{z}}^n \leftarrow \hat{\mathbf{z}}^n \cup \{\hat{z}_q^n\}$ and trusts $\mathbf{r}^n \leftarrow \mathbf{r}^n \cup \{r_q\}$.

In a final step, the third function, $\mathcal{C}$, updates the parameters $\hat{\boldsymbol{\theta}} = \{\hat{\boldsymbol{\theta}}_{(k)}\}_{k=1}^K$ for all experts in the coalition based on $\zeta$, a record of the experts' past labeling decisions from step 1 to $n$.

### B. Implementation of Abstract Functions

As previously mentioned, our backbone includes three abstract functions $\mathcal{A}$, $\mathcal{B}$, and $\mathcal{C}$. The following subsections detail the implementations used in this work.

*1) Expert Ranking and Trust Computation:* The function $\mathcal{A}$ both ranks the experts and calculates their trusts $\{r_q\}_{q=1}^K$. Trust $r_q$ is an estimate of the expert's accuracy, derived through Bayesian inference. Specifically, it is computed as the mean of a $\text{Beta}(s_q + 1, t_q - s_q + 1)$ distribution, which is the posterior distribution of a uniform Beta prior updated with $t_q$ Bernoulli trials. The uniform prior reflects the lack of prior knowledge on expert reliability, which is tied to property (II) from Section II.

Expert ranking is accomplished through a Multi-Armed Bandit (MAB) framework, where each expert is treated as an arm that can yield either a reward of 0 (estimated incorrect answer) or 1 (estimated correct answer). MAB algorithms typically involve a trade-off between exploration (querying

**Algorithm 1** Adaptive multi-expert labels inference

---

**Require:** $S_K$, a coalition of $K$ experts; $\boldsymbol{X}$, a stream of $N$ data points; $\tau \in [0,1]$, a confidence threshold; $\hat{\boldsymbol{\theta}}$, a prior on the experts parameters; and $\mathcal{A}, \mathcal{B}, \mathcal{C}$, three functions to insert to the backbone.

1: Initialize the set of coalitional labels: $\hat{\boldsymbol{Z}}_* \leftarrow \emptyset$
2: Initialize the experts annotations history: $\boldsymbol{\zeta} \leftarrow \emptyset$
3: **for** $n = 1, ..., N$ **do**
4:     $\left(\{a_q\}_{q=1}^K, \{r_q\}_{q=1}^K\right) \leftarrow \mathcal{A}\left(\hat{\boldsymbol{\theta}}, S_K\right)$
5:     Draw next data point $x^n$ from $\boldsymbol{X}$
6:     Obtain estimated label $\hat{z}_1^n$ from expert $a_1$ on $x^n$
7:     $\mathbf{z}^n \leftarrow \{\hat{z}_1^n\}$
8:     $\mathbf{r}^n \leftarrow \{r_1\}$
9:     $c_*^n \leftarrow 0$
10:    $q \leftarrow 1$
11:    **while** $c_*^n < \tau$ **and** $q < K$ **do**
12:       $q \leftarrow q + 1$
13:       Obtain estimated label $\hat{z}_q^n$ from expert $a_q$ on $x^n$
14:       $\mathbf{z}^n \leftarrow \mathbf{z}^n \cup \{\hat{z}_q^n\}$
15:       $\mathbf{r}^n \leftarrow \mathbf{r}^n \cup \{r_q\}$
16:       $(\hat{z}_*^n, c_*^n) \leftarrow \mathcal{B}\left(\mathbf{r}^n, \hat{\mathbf{z}}^n\right)$
17:    **end while**
18:    Append $\hat{z}_*^n$ to $\hat{\boldsymbol{Z}}_*$
19:    Append $\hat{\mathbf{z}}^n$ to $\boldsymbol{\zeta}$
20:    $\hat{\boldsymbol{\theta}} \leftarrow \mathcal{C}(\boldsymbol{\zeta})$
21: **end for**
22: **return** $\hat{\boldsymbol{Z}}_*, \hat{\boldsymbol{\theta}}$

---

less-tested experts to discover their reliability) and exploitation (prioritizing experts with a high estimated $r_q$).

Three selection algorithms with different exploration-exploitation tradeoffs are proposed for ranking experts:

1) **Awake Upper Estimated Reward (AUER) [24]:** This method ranks experts according to an upper bound on their trust estimate, which tightens as they are queried more frequently. This method provides a structured exploration mechanism, with experts being prioritized if their trust is uncertain or high.

2) **Greedy Sampling:** This approach ranks experts purely based on their estimated trust $r_q$, with the exception of unqueried experts, which are prioritized. Although this method may underperform in single-query MAB scenarios due to limited exploration, our problem setting inherently allows multiple experts to be selected per step, which can help compensate for reduced exploration.

3) **Random Sampling:** This explore-only strategy randomly ranks the experts, ensuring that each expert is equally likely to be queried. While less efficient, this approach can be useful in settings where equal data annotation is needed across experts of varying skill levels. This situation typically arise in mixed groups of medical students and professional clinicians, who exploit diagnosis as a platform for continuing medical education

through implicit peer-reviewing processes [4], or in cases where the workload must be evenly distributed.

*2) Coalitional Label Inference:* The function $\mathcal{B}$ computes a coalitional label $\hat{z}_*^n$ for a given data point at step $n$ based on the estimated labels $\hat{\mathbf{z}}^n$ provided by the $q$ experts consulted so far, along with their respective trust scores $\mathbf{r}^n$. The function also yields a confidence score $c_*^n$ that this label is correct.

Assuming independent experts and independent errors across data points for each expert, the probability of any given label $z$ being the true label $z^n$ conditioned on the observed expert labels $\hat{\mathbf{z}}^n$, is proportional to:

$$P(z = z_*^n | \hat{\mathbf{z}}^n) \propto \prod_{i=1}^q \left[ P(\hat{z}_i^n = z^n) \mathbb{1}_{z=\hat{z}_i^n} + P(\hat{z}_i^n \neq z^n) \mathbb{1}_{z \neq \hat{z}_i^n} \right]. \quad (1)$$

Since the trust $r_i$ in each expert $a_i$ estimates the expert's accuracy, the probability $P(\hat{z}_i^n = z^n)$ is approximated by $r_i$, assuming the expert has consistent true positive rate across all classes. Additionally, if we assume that an incorrect label choice is uniformly random across other classes, we estimate:

$$\hat{P}(z = z_*^n | \hat{\mathbf{z}}^n) \propto \prod_{i=1}^q \left[ r_i \mathbb{1}_{z=\hat{z}_i^n} + \frac{1-r_i}{|\mathcal{Z}|-1} \mathbb{1}_{z \neq \hat{z}_i^n} \right], \quad (2)$$

where $\mathcal{Z}$ is the set of possible classes.

To compute the likelihood of $z$ being the true label $z_*^n$ we normalize the right hand side of (2) across all potential labels in $\mathcal{Z}$. The coalitional label $\hat{z}_*^n$ is the one with the highest likelihood, and the associated confidence $c_*^n$ is the corresponding likelihood, given by:

$$c_*^n = \max_{z \in \mathcal{Z}} \frac{\prod_{i=1}^q \left[ r_i \mathbb{1}_{z=\hat{z}_i^n} + \frac{1-r_i}{|\mathcal{Z}|-1} \mathbb{1}_{z \neq \hat{z}_i^n} \right]}{\sum_{z' \in \mathcal{Z}} \prod_{i=1}^q \left[ r_i \mathbb{1}_{z'=\hat{z}_i^n} + \frac{1-r_i}{|\mathcal{Z}|-1} \mathbb{1}_{z' \neq \hat{z}_i^n} \right]}. \quad (3)$$

*3) Inference on the Experts' Parameters:* The function $\mathcal{C}$, which estimates the experts' parameters $\hat{\boldsymbol{\theta}} = \{t_{(k)}, s_{(k)}\}_{k=1}^K$ from the history of their submitted labels $\boldsymbol{\zeta}$, is implemented via the Expectation-Maximization (EM) algorithm [22]. This approach is widely applied in the literature for ground truth, and allows to effectively downweight biased or inconsistent annotators [10], [11], [17]. Since label decisions must be made in real-time, EM is used solely to retrospectively estimate the experts' parameters. For the same reasons as with function $\mathcal{A}$, we employ a uniform Beta prior.

## V. EXPERIMENTAL SETTINGS

This section details the benchmarking process of our algorithm, employing three distinct multi-annotator datasets.

### A. Datasets

We conduct our experiments on one artificial dataset and two publicly available datasets annotated by human respondents. These datasets widely differ in data modality, experts number and abstention rates, and class imbalance, as shown in Table II.

TABLE II
OVERVIEW OF THE DATASETS USED IN THE STUDY. FROM LEFT TO RIGHT, THE COLUMNS REPRESENT THE DATASET NAMES, NUMBER OF EXPERTS,
TOTAL NUMBER OF SAMPLES, NUMBER OF ANSWERS PER DATA POINT, NUMBER OF CLASSES, MODALITY OF $x^n$, ACCURACY BASED ON MAJORITY VOTE,
AVERAGE ACCURACY OF EXPERTS, PROPORTION OF ABSTENTIONS, CLASS IMBALANCE RATIO, AND WHETHER THE EXPERTS ARE ARTIFICIAL OR NOT.

| Dataset | $K$ | $N$ | #answers /data | $|\mathcal{Z}|$ | Data Modality | Accuracy (Majority Vote) | Average Expert Accuracy | Share of Abstentions | Imbalance Ratio | Artificial Experts |
|---|---|---|---|---|---|---|---|---|---|---|
| Glioma Classification (GC) | 6 | 34 406 | 6 | 5 | Images | 0.7465 | 0.6947 | 0.0000 | 71.82 | Yes |
| Weather Sentiment (WS) | 110 | 291 | [13, 20] | 4 | Text | 0.8729 | 0.7140 | 0.8264 | 1.61 | No |
| Music Genre (MG) | 44 | 700 | [1, 7] | 10 | Time Series | 0.7014 | 0.7328 | 0.9044 | 1.15 | No |

*1) Glioma Classification (GC):* This dataset is artificially annotated by six deep neural networks emulating medical experts. It contains histopathological images of brain tumors from the Digital Brain Tumor Atlas [25]. To emulate the varying expertise of medical practitioners, we select models pre-trained on general and specialized datasets. These models classified 34 406 digitized histopathological patches into one of five classes of adult glioma.

*2) Weather Sentiment (WS):* This dataset was gathered using the Amazon Mechanical Turk (AMT)[2], an online crowd-sourcing platform. It contains 291 weather-related tweets divided into 4 classes, each labeled by 13 to 20 out of 110 available experts [26], which reflects staffing levels in most hospital departments. The activity of the experts vary significantly; the most active expert contributed 272 annotations and the least active provided just 1 annotation. The experts accuracy also varies, with a share of the workers even performing worse than random.

*3) Music Genre (MG):* This dataset contains 700 thirty-second music segments classified into 10 genres and annotated by 44 AMT workers, with varying, sometimes adversarial accuracies. Each segment received between 1 and 7 annotations [11]. The activity level of experts in this dataset also varies widely, ranging from 2 to 368 annotations.

In the WS and MG datasets, not all samples are labeled by all experts (see *Share of Abstentions* in Table II). When an expert provides no annotation, the task is rerouted to the next available expert at no additional cost, simulating unavailability. In practical settings, such abstentions can be triggered by simple gatekeeping rules (*e.g.*, fixed time windows to accept and complete tasks), assuming timely responses.

### B. Benchmarked Algorithms

To evaluate the impact of experts allocation based on case difficulty, we compare our algorithm against a baseline that lacks this property. This baseline mirrors our adaptive algorithm, except it selects a fixed number of experts $Q^n = Q \; \forall n$, regardless of the computed confidence score $c_*^n$.

In Section IV, we introduced three methods for selecting the next expert to annotate a data point: AUER, Greedy, and Random. This leads to three variants of both the adaptive and baseline algorithms, resulting in a total of six algorithms for comparison.

[2]https://www.mturk.com/

### C. Evaluation of Algorithms Performances

We use bootstrapping to generate 100 streams from the datasets. For the WS and MG dataset, each stream consists of two-thirds of the data, randomly sampled and ordered. This yields streams of 194 and 466 samples, respectively. Due to the larger volume of the GC dataset, we randomly sample and order 500 points (1.45% of all images) for each iteration.

Each stream is processed by the three variants of the baseline and adaptive algorithms. For the baseline algorithms, we evaluate values of $Q$—the number of experts queried per sample—from 1 to the maximum number of available experts. For the adaptive algorithms, we evaluate confidence thresholds in $\{0.1, 0.2, ..., 0.9\} \cup \{1 - 10^{-n}\}_{n=2}^{15}$. An additional $\tau$ value of 1.1 is included to evaluate the algorithm's performance when the confidence goal is unattainable. For a given value of $\tau$, the whole stream of data was processed in $33.1 \pm 3.8$ s, $11.2 \pm 2.1$ s, and $47.9 \pm 1.5$ s for the GC, WS, and MG datasets respectively, which is negligible compared to the annotation times of the experts.

## VI. RESULTS AND DISCUSSION

This section presents the evaluation of our adaptive algorithm, comparing its performance to the non-adaptive baseline, followed by a discussion on the study's limitations.

### A. Performance Comparison

Figure 2 shows the trade-off between accuracy and the number of queried experts $Q^n$. Each point corresponds to a threshold value defined in Section V-C. For adaptive algorithms, higher confidence thresholds $\tau$ increase $Q^n$ and tend to improve accuracy. However, querying more experts may introduce noise from less reliable annotators, slightly reducing performance at high $Q^n$.

Overall, adapting the number of queries to the latent difficulty of data points yields comparable accuracy at a significantly lower annotation cost. This gain is especially notable when $Q^n$ is neither close to 1 nor $K$, as the algorithm has more room to choose the number of experts. For instance, on the WS dataset, achieving 0.88 accuracy requires 12 experts with the non-adaptive method, but only 6 on average with the adaptive one.

For the GC dataset, Greedy expert sampling performs best. As discussed in Section IV-B1, this is due to multiple expert querying per round, which enables passive exploration despite the strategy being exploit-driven.

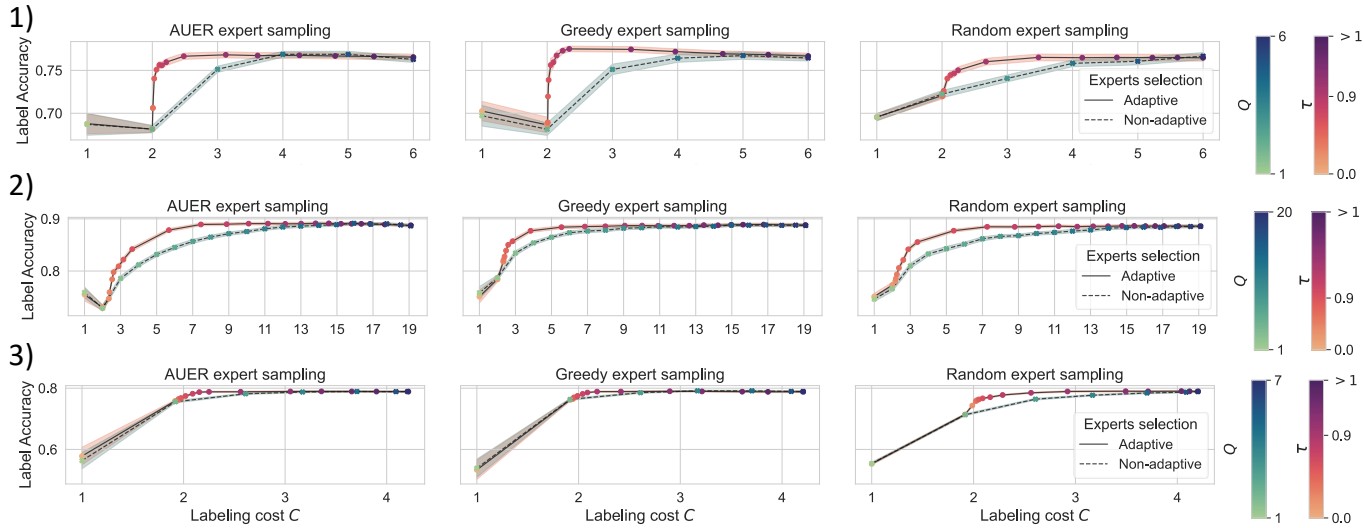

Fig. 2. Comparison of the accuracies of the adaptive and baseline algorithms across different expert sampling strategies for the (1) Glioma Classification, (2) Weather Sentiment, and (3) Music Genre datasets, based on the number of queried experts. Points on the curves represent average performance for a given threshold over 100 bootstrap repetitions. The $\tau$ and $Q$ scales values are indicated in Section V-C. The shaded areas represent the 95% confidence intervals.

TABLE III
GINI COEFFICIENT ON THE DISTRIBUTION OF EXPERT QUERIES ON THE THREE DATASETS. STANDARD DEVIATIONS ARE AFTER THE $\pm$ SIGNS.

| $\tau$ | AUER | Greedy | Random |
|---|---|---|---|
| | (a) Glioma Classification | | |
| 0.4 | $0.0904 \pm 0.0515$ | $0.6619 \pm 0.0020$ | $0.0332 \pm 0.0107$ |
| 0.8 | $0.1339 \pm 0.0290$ | $0.6141 \pm 0.0400$ | $0.0311 \pm 0.0115$ |
| | (b) Weather Sentiment | | |
| 0.4 | $0.1973 \pm 0.0116$ | $0.7356 \pm 0.0125$ | $0.6057 \pm 0.0169$ |
| 0.8 | $0.2210 \pm 0.0103$ | $0.7472 \pm 0.0107$ | $0.5999 \pm 0.0162$ |
| | (c) Music Genre | | |
| 0.4 | $0.5618 \pm 0.0090$ | $0.7798 \pm 0.0235$ | $0.7258 \pm 0.0093$ |
| 0.8 | $0.5738 \pm 0.0080$ | $0.7691 \pm 0.0154$ | $0.7178 \pm 0.0075$ |

### B. Expert Workload Distribution

Table III reports the average Gini coefficient of the query distribution at the end of each dataset stream, reflecting workload disparity among experts (0: uniform load, 1: highly unequal). As expected, Greedy sampling yields the highest inequality across datasets. Interestingly, Random sampling also shows imbalance in WS and MG due to uneven expert availability, as uniform querying maintains abstention disparities over time. In contrast, AUER achieves better balance in these settings, as its exploratory behavior increases the chance of querying rarely available experts when they do appear.

### C. Experts Allocation through Time

An analysis of how the average number of queried experts evolves with the number of annotated data points highlights the adaptive nature of our algorithm. As shown in Fig. 3, the algorithm initially queries many experts when coalition parameters are still uncertain. As it accumulates information,

the number of queried experts quickly drops and stabilizes, demonstrating efficient adaptation to the coalition. This early phase of intense querying offers a practical window to tune the confidence threshold $\tau$. By starting with a high value to promote trust calibration, the user can observe when the system reaches a steady state, and then iteratively adjust $\tau$, *e.g.*, via binary search, to meet desired tradeoffs between annotation cost and accuracy.

### D. Limitations and Future Directions

While our results demonstrate the effectiveness of adaptive expert querying for classification tasks, which are common in medical AI workflows, future work will extend our modular framework to structured prediction and ordinal tasks. For example, pixel-wise aggregation and Dice-based expert ranking could support segmentation, while EM updates could be adjusted to reflect ordinal distances.

Due to the lack of large-scale clinician-annotated datasets with ground truth, we evaluated our method across three diverse datasets (synthetic and real, with varying expert availability and class imbalance). Despite this constraint, we observed consistent convergence behavior (Figs. 2, 3), supporting our framework's generalizability from label-only input. Future work will however focus on clinical validation.

Finally, while expert reliability was modeled as static due to the short sampling period of available datasets, our design accommodates evolving expertise via simple modifications to $\mathcal{C}$ (*e.g.*, sliding windows or exponential decay).

Overall, the flexibility of our framework and its modular backbone set a solid premise to explore these extensions in future works.

## VII. CONCLUSION

In this paper, we introduced an adaptive ground truth inference algorithm designed to integrate medical screening

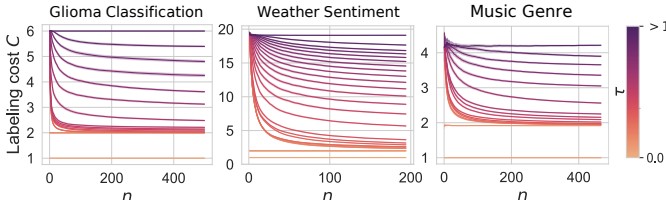

Fig. 3. Average number of queried experts through time for the adaptive algorithm with AUER expert sampling over 100 bootstrap repetitions. Similar curves are observed with the Greedy and Random expert samplings. The shaded areas represent the 95% confidence intervals.

pipelines by aligning with medical practices. This algorithm is capable of jointly (I) providing on-the-fly annotations for unseen data points, (II) deploying on a coalition of unknown experts without requiring external information, and (III) allocating more experts resources to data points that are more difficult to label. Our experiments have demonstrated that this third property significantly reduces cost for a similar accuracy target compared to algorithms that distribute experts resources evenly across all data points.

By offering an algorithm integrating modular abstract functions, we aim to stimulate further research in the field of medical ground truth inference.

## ACKNOWLEDGMENTS

T. Bary is funded by the Walloon region under grant No. 2010235 (ARIAC). T. Godelaine is supported by MedReSyst, funded by the Walloon Region and EU-Wallonie 2021-2027 program. A. Abels is a Postdoctoral Researcher (CR) of the Fonds de la Recherche Scientifique – FNRS. Computational resources have been provided by the CÉCI, funded by the F.R.S.-FNRS under Grant No. 2.5020.11 and the Walloon Region.

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
