# OpenReview forum: "Optimizing Resources for On-the-Fly Label Estimation with Multiple Unknown Medical Experts"
_IEEE.org/EMBS/BHI/2025/Conference — BHI 2025_

### Official Review · Reviewer_Aprs · 2025-07-01
**Optimizing Resources for On-the-Fly Label  Estimation with Multiple Unknown Medical Experts**

**Confidence:** 5
**Clarity Of Writing:** great
**Clinical Significance:** good
**Methodological Novelty:** good
**Overall Rating:** 5

**Experiments And Results:**

good

**Questions For The Authors:**

- Can the trust model adapt to evolving expert proficiency (e.g., new trainees or specialists gaining experience)? if you agree this could boost the robustness in educational or workforce rotation, try it.
Bcz, - I'm concern about trust reversion, for instance if an expert improves or degrades over time, how does the system handle dynamic recalibration?

- How was the confidence threshold T chosen in practical?does it depends on cost benefits tradeoffs?
- Is it -possible to include per-expert load balancing metrics (e.g., Gini coefficient of query distribution) to speak to fairness and cognitive burden?
- Also it might be worth comparing against RL-based annotator selection frameworks to contextualize gains from the MAB strategies used
- In Section IV-B1, exploring the impact of alternative priors (e.g., informative Beta) might refine early-stage trust estimates

**Strengths:**

- It alighns with standard world medical screening practices workflow.
- Use of Bayesian trust estimation and MAB-based expert sampling, is great especially under high uncertaininty
- Clear abstraction of core functions (ranking, inference, parameter update) promotes extensibility.
- Fig. 3 clearly illustrates the system's ability to "learn the coalition" over time, and that’s a key strength for scalable deployment.

**Summary Of The Paper:**

This paper proposes a stream-based adaptive algorithm for aggregating expert annotations in medical screening workflows, particularly when expert proficiency is unknown and data arrives continuously. The Paper states, it implemented a modular algorithm that queries experts incrementally using a multi-armed bandit strategy (AUER, Greedy, or Random sampling). A Bayesian trust estimation is used to rank experts, and a coalitional label is generated through probabilistic inference over trusted annotations. The model is evaluated on three multi-annotator datasets, glioma classification, weather sentiment, and music genre, and shows up to 50% reduction in annotation cost while maintaining label accuracy comparable to fixed-query baselines. The method aims to:
•	Provide on-the-fly annotations without requiring pre-labeled data or known expert reliabilities.
•	Dynamically allocate annotation resources (expert queries) based on instance difficulty.
•	Use confidence-based stopping criteria to determine when enough expert input has been gathered.

**Weaknesses:**

- Assumes consistent expert performance; real-world expertise may evolve over time (e.g., fatigue, learning).
- In datasets with high abstention rates, rerouting may not reflect true delays or decision bottlenecks in clinical settings

---

### Official Review · Reviewer_ZjD1 · 2025-07-12
**Optimizing Resources for On-the-Fly Label Estimation with Multiple Unknown Medical Experts**

**Confidence:** 4
**Clarity Of Writing:** great
**Clinical Significance:** good
**Methodological Novelty:** fair
**Overall Rating:** 6

**Experiments And Results:**

good

**Questions For The Authors:**

How does the proposed method compare to recent probabilistic models for label aggregation under uncertainty, such as Bayesian approaches or deep ensemble methods? Would these offer complementary or superior capabilities for dynamic expert selection?

Have the authors considered extending their framework to tasks beyond classification, such as medical image segmentation or ordinal grading? What challenges would arise in adapting the current backbone to these more complex annotation types?

How does the method handle evolving expert reliability over time, as would occur in real clinical environments where practitioners gain experience or shift roles?

What are the computational requirements of the adaptive algorithm in a real-time setting? Could the confidence updates and multi-armed bandit ranking introduce latency that would hinder deployment in high-throughput screening?

Clear answers to these questions could strengthen confidence in both the novelty and practical applicability of the proposed system.

**Strengths:**

The paper addresses an important and highly practical challenge in medical AI workflows: how to efficiently aggregate noisy and heterogeneous expert annotations in settings where data arrives as a continuous stream. This reflects real-world screening programs where timely and accurate decisions are critical. The authors demonstrate a clear understanding of the constraints of such environments, particularly the scarcity of labeled data and the need for algorithms that can “cold-start” with no prior information on expert reliability.

A key strength of this work lies in its adaptive resource allocation strategy. By integrating a confidence-driven stopping criterion with trust-weighted voting, the proposed method allows difficult or ambiguous cases to receive greater expert attention while reducing redundancy on easier cases. This approach is both conceptually elegant and computationally efficient, showing measurable reductions in annotation overhead without sacrificing label quality.

The use of a multi-armed bandit framework for expert ranking is particularly noteworthy. It provides a principled way to balance exploration and exploitation when selecting annotators, which is critical in medical settings where some experts may have highly specialized skills. The modular design of the algorithm, with abstract functions for ranking, inference, and parameter updates, also enhances its adaptability to diverse annotation tasks beyond classification.

The experimental setup is comprehensive and includes both artificial and human-annotated datasets. The authors provide clear visualizations of trade-offs between labeling cost and accuracy and analyze the temporal dynamics of expert allocation, offering insights into how their method adapts as more data flows into the system.

**Summary Of The Paper:**

This paper proposes a novel adaptive algorithm for real-time ground truth estimation in medical screening scenarios where annotations are provided by multiple experts of initially unknown reliability. The method is designed to support on-the-fly labeling of continuously incoming data without requiring prior knowledge of expert proficiency or pre-labeled datasets. At its core, the algorithm incrementally queries additional experts for a given data point until a predefined confidence threshold is reached. This adaptive querying strategy aims to balance annotation accuracy and resource allocation by dynamically focusing expert attention on more challenging cases.

The system is implemented as a modular backbone consisting of three abstract functions: (A) expert ranking and trust computation using a multi-armed bandit framework, (B) coalitional label inference with confidence estimation, and (C) dynamic updates of expert parameters through an Expectation-Maximization (EM) approach. The method is evaluated on three multi-annotator classification datasets (histopathological glioma images, crowd-annotated weather sentiment, and music genre classification) and demonstrates that adaptive expert querying can reduce annotation costs by up to 50% while maintaining accuracy comparable to a non-adaptive baseline.

**Weaknesses:**

Despite these strengths, the novelty of the proposed method is somewhat incremental when viewed against the broader landscape of label aggregation and active learning. Many elements, such as trust-weighted voting, EM-based parameter updates, and adaptive querying, have precedent in the literature. While the paper cites related works, it would benefit from a clearer articulation of how its approach substantively advances beyond established methods like CLARA (Nguyen et al., KDD, 2020) or Welinder and Perona’s online crowdsourcing framework (CVPRW, 2010).

The reliance on classification tasks for evaluation also limits the generalizability of the findings. Medical annotation often involves more complex tasks such as regression (e.g., tumor grading) or segmentation (e.g., lesion delineation), where adaptive expert allocation could behave differently. Extending the framework to these modalities would be an important next step.

Furthermore, the datasets used in the experiments, while diverse, are relatively small and in some cases synthetic (e.g., glioma classification with pseudo-experts). This raises concerns about how the method would perform in larger-scale, clinically realistic environments with dozens or hundreds of annotators and highly imbalanced classes.

The paper’s discussion of practical deployment is also limited. In real medical screening workflows, annotators are not always available on demand, and their expertise may evolve over time. The current approach assumes static expert skill levels and instantaneous responses, which may oversimplify the dynamics of real-world systems. Finally, while the adaptive querying demonstrates cost savings, no formal analysis of latency or computational overhead is provided, important factors in real-time screening contexts.

---

### Official Review · Reviewer_5fc4 · 2025-07-15
**Promising Approach but Needs Realistic Data and Practical Considerations**

**Confidence:** 3
**Clarity Of Writing:** good
**Clinical Significance:** good
**Methodological Novelty:** good
**Overall Rating:** 4
**Final Rating:** 5

**Experiments And Results:**

good

**Questions For The Authors:**

NA

**Strengths:**

The paper is well written and generally easy to follow.
The proposed method is reasonable and logically developed.

**Summary Of The Paper:**

This paper focuses on improving accurate prediction and estimation in medical screening with the goal of reducing query costs, which is meaningful and relevant in the crowdsourcing domain.

**Weaknesses:**

1.	Although the study aims to be applicable in clinical settings, only one dataset is related to the medical domain, and the data is artificially generated. This limits the evidence for real-world applicability.
2.	For the ranking system, the trust value is defined based on the number of queries and the number of correct labels. However, different experts may have varying strengths across subdomains, for example, an expert may perform well in subdomain A but poorly in subdomain B. In this case, it may be more appropriate to compute trust scores domain by domain rather than using a single global score.
3.	The framework optimizes the allocation of expert resources in medical screening by ranking experts according to their trust levels and selecting the minimum number needed to meet a predefined threshold. This is reasonable from a cost perspective. However, in clinical practice, should cost be the only factor? Junior experts typically have lower trust scores due to limited experience, but they need opportunities to practice and improve. This practical consideration is not discussed and may limit the framework’s real-world feasibility.
4.	In Sections III and IV, the symbol ‘n’ is overused and overloaded with different meanings, which could cause confusion. Clearer notation or more precise definitions are needed.

---

### Official Review · Reviewer_QUsh · 2025-07-15
**An efficient and adaptive annotation aggregation method**

**Confidence:** 4
**Clarity Of Writing:** good
**Clinical Significance:** great
**Methodological Novelty:** good
**Overall Rating:** 7

**Experiments And Results:**

great

**Questions For The Authors:**

- How would the framework perform in the presence of adversarial or biased annotators?
- Can the framework be extended to structured prediction problems such as segmentation?

**Strengths:**

- The paper addresses an impactful real-world problem, as medical datasets are costly to label.
- The paper performs extensive evaluation on three different datasets from three different domains, which suggests generalizability of the proposed method.
- The related work section is well written, giving the reader a comprehensive view of literature in this area.

**Summary Of The Paper:**

This paper addresses the challenge of label estimation in clinical machine learning settings where annotations are collected from multiple medical experts of unknown reliability. The paper proposes an adaptive labeling framework that can aggregate annotations from multiple experts real-time using streaming data, does not require prior information on medical experts, and can adaptively seek more annotations from experts for a sample. The proposed method is evaluated on three datasets and is compared against the baseline which is identical to the proposed algorithm in all aspects except that the baseline queries a fixed number of experts.

**Weaknesses:**

- The model appears to assume that annotator errors are independent across data points, which may not hold in practice, as they may sometimes be influenced by specific case types or visual patterns. It would be good to have a clarification in the paper on whether this assumption is being made.
- Scalability to larger expert annotator pools is not discussed in the paper: how does the proposed method scale when the number of available expert annotators increases to 100/1000/10000/... etc?
- The paper does not offer comparisons with some relevant literature [1] and [2]. It would be good to have a quantitative or at least qualitative understanding of how the proposed method compares to these works.

References
- [1] S. Paun and E. Simpson. 2021. Aggregating and Learning from Multiple Annotators. In Proceedings of the 16th Conference of the European Chapter of the Association for Computational Linguistics. https://aclanthology.org/2021.eacl-tutorials.2.pdf
- [2] D. Mahapatra, “Combining multiple expert annotations using semi-supervised learning and graph cuts for medical image segmentation,” Computer Vision and Image Understanding, vol. 151, pp. 114–123, Oct. 2016. https://doi.org/10.1016/j.cviu.2016.01.006.
‌

---

### Official Review · Reviewer_r2Az · 2025-07-18
**An adaptive expert annotation tool that reduces labeling effort while preserving accuracy.**

**Confidence:** 3
**Clarity Of Writing:** great
**Clinical Significance:** excellent
**Methodological Novelty:** great
**Overall Rating:** 7

**Experiments And Results:**

great

**Questions For The Authors:**

1. The authors could extend the framework by introducing per-expert cost modeling or query quotas, and optimize a global cost function that accounts for both accuracy and workload fairness such as adding expert-specific query cost terms.
2. The choice of tau is critical for the cost-accuracy tradeoff. What is your suggestion for the real-world deployment.
3. Is there any plan for real world clinical expert validation?

**Strengths:**

1. The method is designed for real-world medical annotation pipelines, where clinician time is limited and labeling cost is high.
2. The use of Bayesian method, adaptive querying are rigorous and effective.
3. Substantial reduction in expert queries with no loss in accuracy, as shown across three datasets.

**Summary Of The Paper:**

This paper proposes an adaptive ground truth inference algorithm for real-time labeling with multiple unknown experts. The method dynamically queries experts based on instance difficulty and expert trust scores, using a framework containing three components: expert ranking, label aggregation, and trust updating. Evaluated on three multi-annotator datasets, the adaptive approach reduces labeling cost (number of expert queries) by up to 50% while maintaining comparable accuracy to non-adaptive baselines. I think it will be a generic useful tool for prospectively annotating the data for the clinicians that could potentially reducing their effort.

**Weaknesses:**

1. While the method is positioned for clinical use, none of the datasets involve annotations from real clinicians. The Glioma dataset uses synthetic experts (deep learning models), while the other two datasets are crowdsourced and non-medical. This limits the ability to assess real-world feasibility and expert interaction.

2. A potential practical issue is that the highest-ranked experts (i.e., most trusted) are likely to be queried the most, possibly for nearly every data point. In real-world settings, this could lead to overburdening a small subset of clinicians, defeating the goal of efficient resource allocation.